# Maritime Continent water cycle regulates low-latitude chokepoint of global ocean circulation

Tong Lee [1], Séverine Fournier[1], Arnold L. Gordon [2] & Janet Sprintall[3]

The Maritime Continent (MC) is a low-latitude chokepoint of the world oceans with the Indonesian throughflow (ITF) linking the Indo-Pacific oceans, influencing global ocean circulation, climate, and biogeochemistry. While previous studies suggested that South-China-Sea freshwaters north of the MC intruding the Indonesian Seas weaken the ITF during boreal winter, the impact of the MC water cycle on the ITF has not been investigated. Here we use ocean-atmosphere-land satellite observations to reveal the dominant contribution of the MC monsoonal water cycle to boreal winter−spring freshening in the Java Sea through local precipitation and runoff from Kalimantan, Indonesia. We further demonstrate that the freshening corresponds to a reduced southward pressure gradient that would weaken the ITF. Therefore, the MC water cycle plays a critical role regulating ITF seasonality. The findings have strong implications to longer-term variations of the ITF associated with the variability and change of Indo-Pacific climate and MC water cycle.

[1] Jet Propulsion Laboratory, California Institute of Technology, Pasadena, CA 91011, USA. [2] Lamont-Doherty Earth Observatory, Columbia University, Palisades, NY 10964, USA. [3] Scripps Institution of Oceanography, University of California, San Diego, CA 92037, USA. Correspondence and requests for materials should be addressed to T.L. (email: tlee@jpl.nasa.gov)

The Indonesian Archipelago, a maritime chokepoint of global ocean circulation, provides the only low-latitude connection of the world oceans. The Indonesian throughflow (ITF) passes through different basins and narrow straits of the Maritime Continent (MC) region, linking the tropical Pacific and Indian Ocean circulations (Fig. 1). A vast body of literature has discussed the importance of the ITF to ocean circulation physics, climate variability, and biogeochemistry[1–9]. More recent studies also implicated the ITF's role in sea level changes of the southeast Indian Ocean[10,11] and in the basin-scale oceanic heat content variability associated with the so-called global warming hiatus[12].

The ITF is influenced by remote forcing from both the Pacific and Indian Oceans[13–15]. In addition, regional circulations within the greater Southeast Asian Sea (SEAS) region, which includes the Maritime Continent in its southern part, also influence the ITF. In particular, the South China Sea (SCS) throughflow is believed to affect the ITF vertical structure and temporal variability[16–23]. The relatively fresh SCS surface-layer waters advected by the monsoonal wind-driven ocean currents through the shallow (<40 m) Karimata Strait into the Java Sea in the MC during boreal winter was suggested to create an upper-layer freshwater plug that spreads into the Makassar Strait[18]. The latter is the main channel of the ITF. The boreal-winter freshwater plug results in an anomalously large dynamic height in the upper layer, which weakens the predominant north-to-south pressure gradient and thus the southward ITF transport in the upper layer[17]. Consequently, the ITF has a subsurface maximum in its velocity profile during boreal winter. On interannual time scales, SCS freshwaters also intrude into the northern Makassar Strait via the Sibutu Passage[18], thereby affecting the vertical profile of the ITF. The modification of the vertical structure of the ITF due to freshwaters has important implications to the transport-weighted exchanges of heat, freshwater, and biogeochemical properties between the Pacific and Indian Oceans[17,24–26].

Although these previous studies have considered the effect of the SCS throughflow and the fresher SCS waters, no effort has been made to investigate the influence of the local monsoonal water cycle of the MC on the seasonal freshwater plug. Our understanding of freshwater changes in the MC region and their relationship with the regional water cycle have been hampered by the paucity of in situ salinity measurements in the region. Moreover, most ocean models offer little prospect for elucidating the mechanism controlling sea surface salinity (SSS) because of the common practice of relaxing model SSS to seasonal SSS climatology to prevent model drift caused by errors in surface forcing and deficiencies in model physics. The SSS relaxation, which is a statistical correction to compensate the errors due to surface forcing and model deficiencies, complicates the investigation of the relationship between SSS and precipitation using model output.

Recent advance in salinity remote sensing provides the capability to fill this knowledge gap. The Soil Moisture and Ocean Salinity (SMOS) satellite launched by the European Space Agency (ESA) has been providing SSS retrievals since 2010 at 43-km spatial resolution[27]. Until very recently, SMOS SSS retrievals have been significantly affected by land signal contamination within 1000 km of the coasts. A newly available SMOS dataset brought significant improvements of data quality in coastal oceans and marginal seas[28]. The Aquarius satellite (2011–2015) launched by the National Aeronautics and Space Administration (NASA) has a 150-km spatial resolution[29]. Although Aquarius SSS have been widely used to study ocean-open SSS variability[30–35], the resolution is too coarse to study SSS variations in much of the MC except for regions such as the SCS[36,37]. NASA's Soil Moisture Active Passive (SMAP) satellite, launched in January 2015, has been providing SSS retrievals at 40-km spatial resolution. SMAP SSS data are much less affected by radio frequency interference (RFI) than SMOS and Aquarius because of SMAP's advanced RFI detector[38], which greatly facilitated the studies of SSS variations in coastal oceans and marginal seas[35,37,39,40]. The simultaneous measurements of SSS and soil moisture by SMAP has benefited the research on land—sea linkages[39]. Therefore, SMAP data present an improved capability to study water-cycle-related changes from sea to land in the MC region.

In this study, we provide the first systematic characterization of the spatial and temporal evolution of the seasonal freshwater plug in the Indonesian Seas by capitalizing on the unprecedented capability of SMAP SSS to monitor marginal sea salinity. We reveal a direct linkage of the source of the freshwater plug with the regional water cycle in the MC using SMAP data and other ocean and atmosphere satellite observations (see Data in the Methods section). Moreover, we demonstrate that the seasonal freshwater plug not only exists in boreal winter, but prolongs into boreal spring. We further illustrate the relationship of the seasonal freshwater plug with the meridional pressure gradient along the Makassar Strait that regulates the ITF using satellite observations. The findings represent a significant advance in understanding the linkage of ocean circulation with the water cycle. The results also have implications to longer-term changes of the ITF associated with the variability and change of Indo-Pacific climate and MC water cycle.

## Results

**Spatiotemporal structure of the seasonal freshwater plug**. Here we document the spatial and temporal variations of the seasonal freshwater plug in the Indonesian Seas. The seasonal composites of SMAP SSS in the SEAS for the study period of April 2015—March 2018 (Fig. 2a–d) show that the SSS in the MC, especially the Java Sea and the Makassar Strait, are fresher during boreal winter (December–January–February, DJF) and boreal spring (May–April–May, MAM) and saltier during boreal summer (June–July–August, JJA) and boreal fall

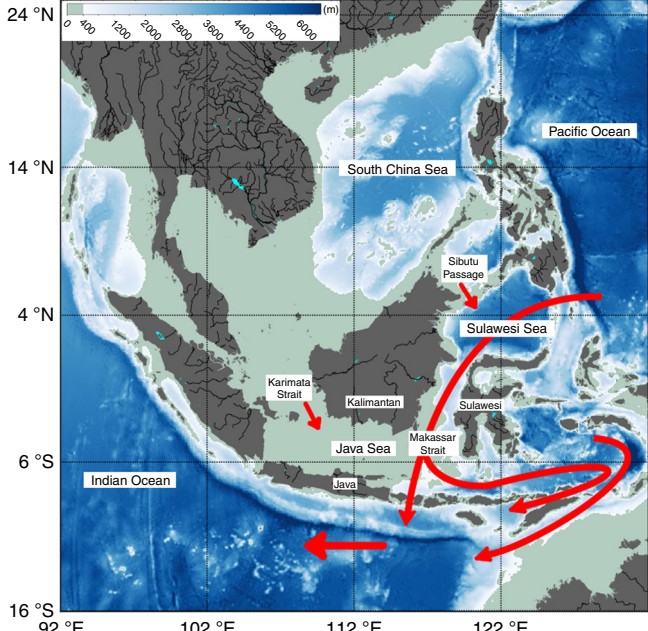

**Fig. 1** Bathymetric map for the southeast Asian Seas and the major pathways of ocean currents in the Maritime Continent region. The schematic illustration is based on refs. [9,17]

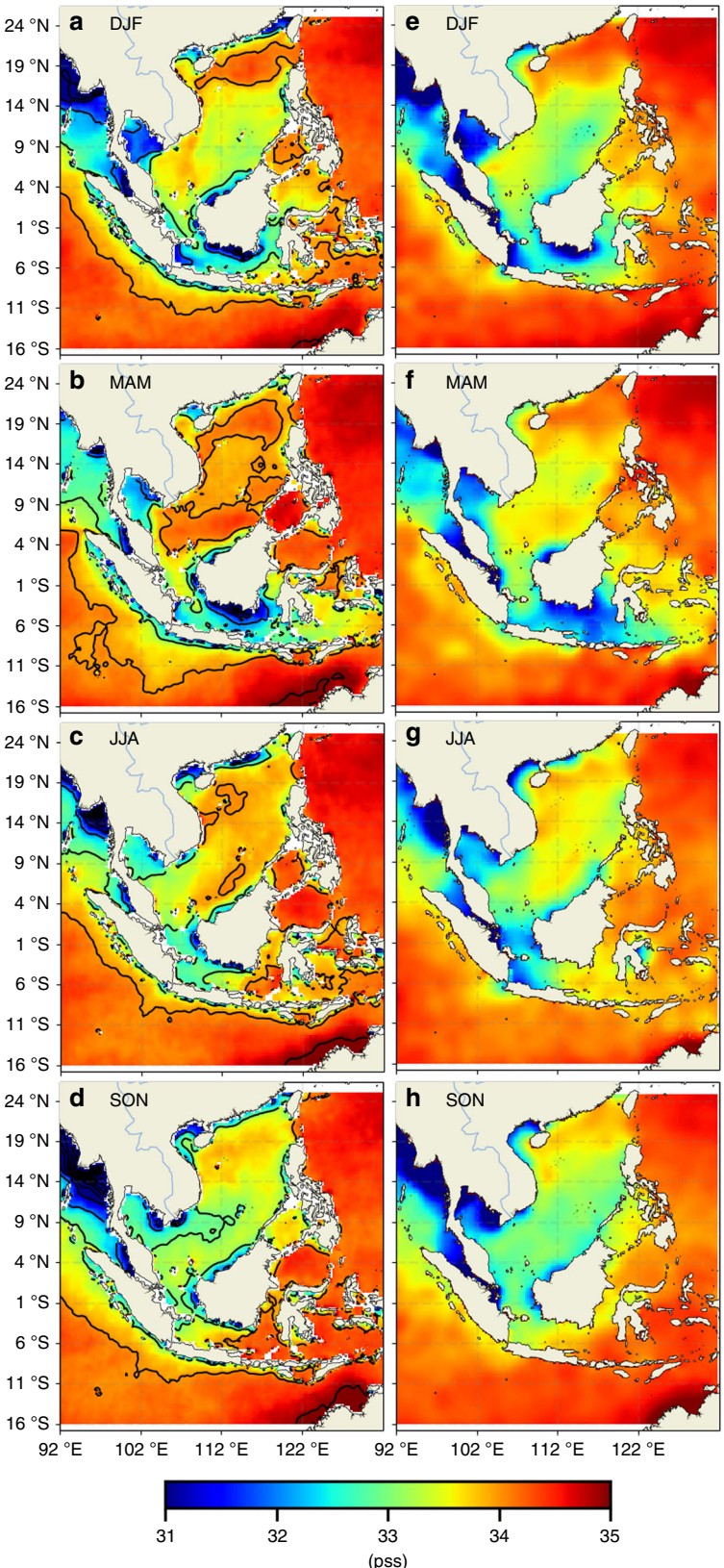

**Fig. 2** Seasonal composites of sea surface salinity from the SMAP satellite and from the World Ocean Atlas 2013 climatology. **a–d** SMAP sea surface salinity composites for the DJF, MAM, JJA, and SON seasons. **e–h** World Ocean Atlas 2013 sea surface salinity climatology for the DJF, MAM, JJA, and SON seasons

(September–October–November, SON). The opposite is generally true for the SSS in the northern SEAS, including the SCS to the north of the MC.

The low SSS during DJF in the Java Sea and Makassar Strait (Fig. 2a) demonstrate, for the first time, the space-based capability to characterize the boreal-winter freshwater plug. The SMAP SSS also suggest that the freshwater plug actually persists into MAM (Fig. 2b), a feature that has not been documented previously. The SSS values in Java Sea and southern Makassar Strait during DJF and MAM are much lower than those in the SCS in any season. This suggests that the SCS freshwaters are not the main source of the freshwater plug, in contrast to what was suggested previously[18].

The extremely sparse in situ salinity measurements in the SEAS region make it impossible to generate seasonal composite maps like the ones from SMAP based on in situ data over the 3-year period of the study. We therefore compare the SMAP SSS seasonal composites (Fig. 2a–d) with the seasonal climatology of SSS from the World Ocean Atlas 2013 (WOA13, see Data in the Methods section) (Fig. 2e–h). The WOA13 climatology was constructed based on historical data, which are quite inhomogeneous in terms of spatiotemporal sampling in the SEAS region. Hence, it is unclear how representative the WOA13 climatology is of the real SSS conditions. However, the major patterns of the seasonal variations of SSS in the SMAP and WOA13 datasets are similar, for instance, both showing the fresher waters in the Java Sea and Makassar Strait and the saltier waters in the SCS during DJF and MAM. The spatial gradients of the WOA13 SSS climatology are not as sharp as those of SMAP SSS. Moreover, the spatiotemporal inhomogeneity of the in situ measurements used to construct the WOA13 climatology appear to have caused some features that are probably unrealistic. During MAM, for instance, the freshening in the eastern Java Sea that appears as a thick blue cross pattern (Fig. 2f) is likely caused by the inhomogeneity of the in situ measurements and the related smoothing in producing the gridded climatology.

The SMAP SSS in the SEAS tend to be saltier than those of WOA13. This is consistent with the fact that the period of the SMAP data (April 2015 onward) coincides with a positive Indian-Ocean Dipole (IOD) event (in the boreal autumn of 2015) and the strong 2015–2016 El Niño event. During positive IOD (El Niño), atmospheric convection and the associated precipitation generally shift westward (eastward) away from the SEAS region because of the anomalously low SST in the eastern tropical Indian Ocean (western tropical Pacific)[41–43]. This can result in anomalously low precipitation in the SEAS region, thereby causing large-scale salinification of the region in general.

To illustrate the relationships of the seasonal SSS anomalies with ocean currents, the composite maps of SMAP SSS anomalies (referenced to April 2015–March 2018 time mean) are superimposed on ocean total surface currents from the HYCOM operational ocean analysis (see Data in the Methods section) (Fig. 3). During DJF, the currents through the Karimata Strait driven by the monsoonal winds carry waters from the SCS into the Java Sea and from the Java Sea up into the Makassar Strait (Fig. 3a). The opposite is true during JJA (Fig. 3c). This seasonally varying pattern of the ocean surface currents is well-known[17–21]. The SCS waters advected through the Karimata Strait into Makassar Strait and the Indonesian Seas by the boreal-winter ocean currents are indeed fresher than the waters advected into the Makassar Strait from the north that originate from the salty Northwest Pacific (Fig. 2a). Therefore, the relatively low salinity waters from the SCS are expected to contribute to the boreal-winter freshwater plug to some extent. However, during DJF and MAM, the SSS north of the Karimata Strait is higher than those in the Java Sea (Fig. 2a–c). The same is true for SSS anomalies

(Fig. 3a, b). Therefore, the SCS waters cannot be the primary source of freshwater for the strong freshening in the Java Sea and Makassar Strait observed during DJF and MAM even though the SCS throughflow has an important dynamical effect in carrying the Java Sea freshwater into the Makassar Strait.

**Relationship with regional water cycle in the MC.** A critical knowledge gap in understanding the cause for the seasonal freshwater plug is the effects of the regional monsoonal water cycle in the MC region, both in terms of regional precipitation and runoff. Here we will investigate these effects. The composite seasonal maps of precipitation anomalies (referenced to the April 2015–March 2018 time mean) (Fig. 4) show that the precipitation anomalies over the Java Sea and Kalimantan (Indonesian part of the Borneo island) are generally positive (negative) during DJF (JJA), vice versa for the SCS. This pattern reflects the different wet and dry seasons associated with the monsoon passage over the MC in the southern SEAS and over the northern SEAS[44–46]. In particular, the precipitation anomalies that extend zonally across the entire Java Sea during DJF are particularly intense (up to approximately 240 mm per month). These precipitation anomalies would reduce the SSS in the Java Sea from DJF to MAM. This is because precipitation causes the temporal change of SSS (i.e., SSS tendency); thus, the seasonal SSS anomaly is in quadrature relation with the seasonal precipitation, with the former lagging the latter by one season. In fact, the values of SSS anomalies over much of the Java Sea decrease from DJF to MAM, consistent in sign with the cumulative effect of the DJF precipitation anomalies over the region. This suggests that the DJF precipitation contributes to the freshwater plug observed during DJF and MAM. The evaporation anomalies are almost an order of magnitude smaller than precipitation anomalies for the Java Sea region (see Method) and play a minor role in regulating the seasonal SSS.

The DJF and MAM SSS (Fig. 2a, b) and SSS anomalies (Fig. 3a, b) are not uniform in the Java Sea and the southern Makassar Strait. They show lower values off the southern and eastern coasts of Kalimantan. This is an indication of the effect of runoff from Kalimantan that re-enforces the freshwater plug and prolongs it from DJF to MAM. While the precipitation anomalies over the Java Sea are positive only during DJF (more rain), those over Kalimantan are positive during both DJF and MAM (Fig. 4a, b), thus prolonging the runoff from Kalimantan from DJF to MAM. Although the precipitation anomalies over Kalimantan are weaker than those over the Java Sea, the effects of river runoff from Kalimantan into the Java Sea are concentrated along the coastal oceans. Therefore, the impact of runoff on SSS per unit area in the coastal oceans off Kalimantan is not necessarily smaller than the effect of local precipitation over the Java Sea. SMAP SSS data within one satellite footprint (40 km) of the coasts have been masked out to avoid the contamination of SSS data by land signals. Were SMAP data within 40km of the coasts reliable, we would expect to see even lower SSS values along the southern and eastern coasts of Kalimantan.

Runoff measurements from Kalimantan are not publicly available. We therefore seek additional supporting evidence for boreal-winter–spring runoff from Kalimantan from other satellite measurements. The first dataset we examined is soil moisture measurements from SMAP (see Data in the Methods section). The composite of seasonal anomalies from SMAP soil moisture show that the soils in Kalimantan are in general anomalously wet during DJF and MAM (Fig. 5a, b), particularly in the southern part of Kalimantan, reflecting the impact of the precipitation anomalies shown in Fig. 4a, b. Another dataset that we used to provide evidence of runoff is colored dissolved and detrital organic matter (CDOM) satellite ocean color measurements (see

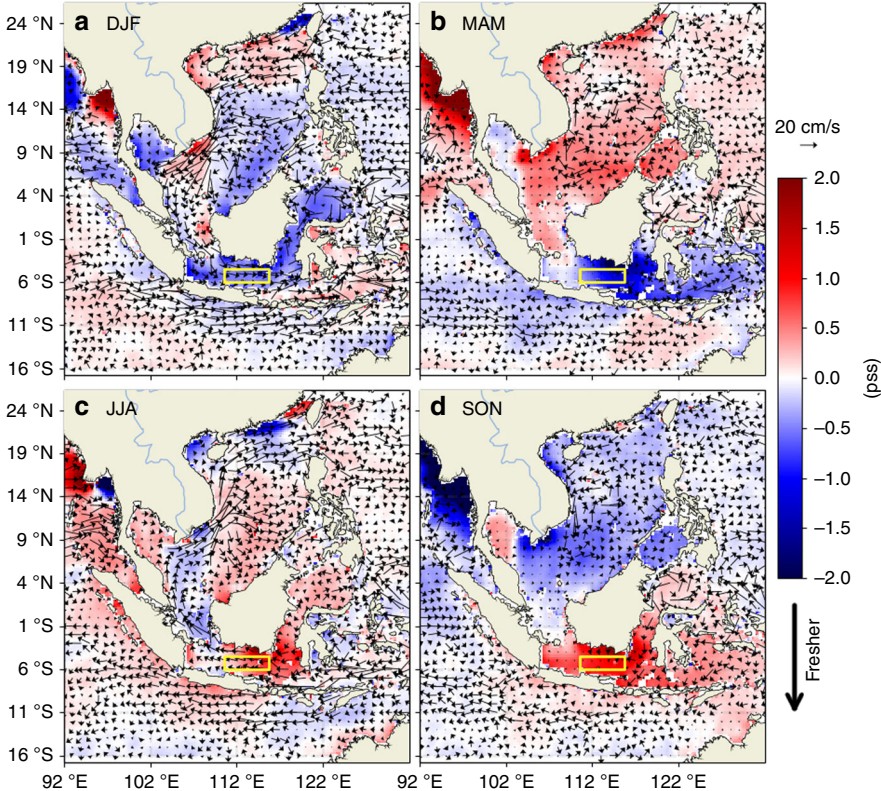

**Fig. 3** Composite seasonal maps of sea surface salinity anomalies from the SMAP satellite with ocean surface currents superimposed. **a–d** correspond to the maps for the DJF, MAM, JJA, and SON seasons. To facilitate visual inspection, the current vectors are plotted every 1°. The yellow box (110.5–115.75 °E, 6–4.5 °S) indicates the region where the sea surface salinity budget analysis was performed (results shown in Figs. 7, 8)

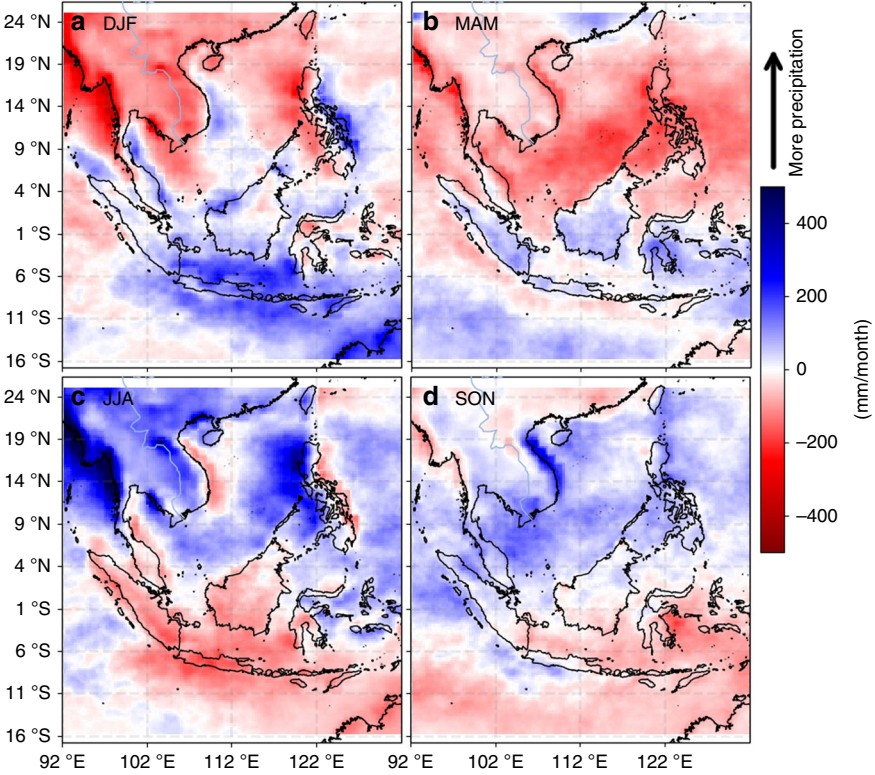

**Fig. 4** Composite seasonal maps of precipitation anomalies. **a–d** correspond to the maps for the DJF, MAM, JJA, and SON seasons

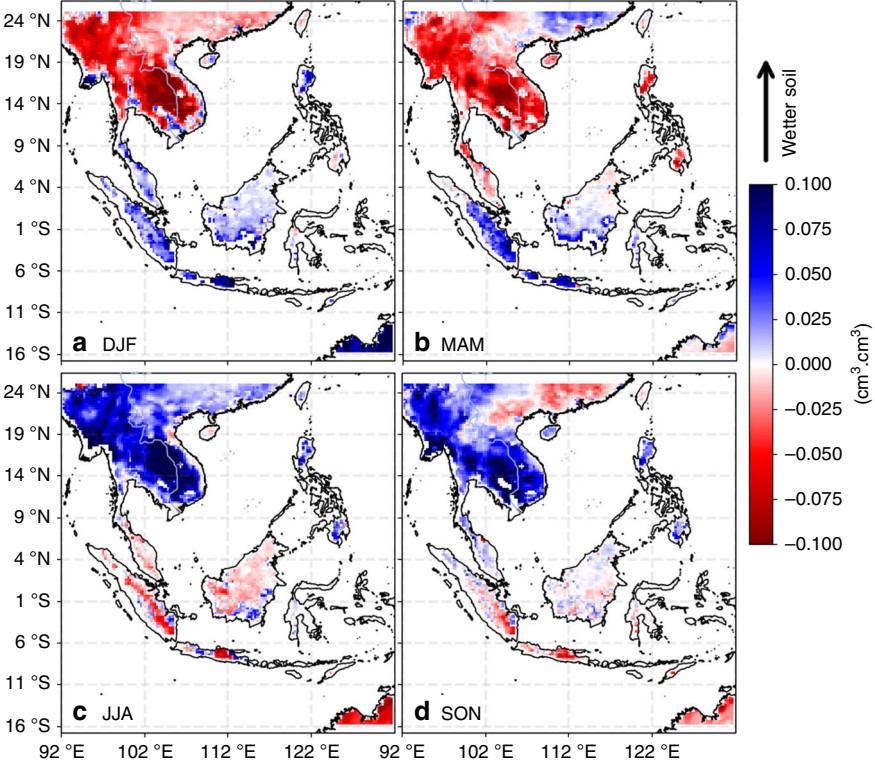

**Fig. 5** Composite seasonal maps of soil moisture anomalies. **a**–**d** correspond to the maps for the DJF, MAM, JJA, and SON seasons

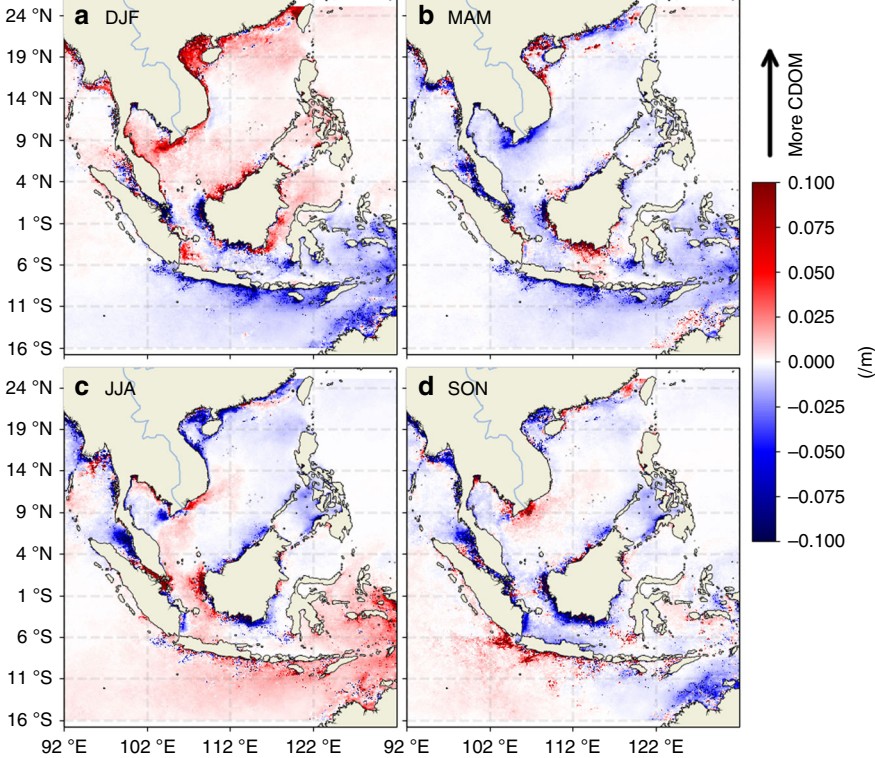

**Fig. 6** Composite seasonal maps of Colored Dissolved Organic Matter anomalies. **a**–**d** correspond to the maps for the DJF, MAM, JJA, and SON seasons

Data in the Methods section). Coincident increase in CDOM and decrease in SSS have been widely considered as an evidence of continental runoff because the runoff brings both freshwater and CDOM[47]. The composite seasonal anomalies of CDOM (Fig. 6) show anomalously high CDOM in the coastal oceans east and south of Kalimantan during DJF and MAM, coinciding with the anomalously low SSS (Fig. 2a, b) and SSS anomalies (Fig. 3a, b). These features reflect the impact of runoff from Kalimantan

caused by the precipitation anomalies over the island during this period. This runoff effect re-enforces and prolongs the seasonal freshwater plug from DJF to MAM.

The discussion above suggests that regional monsoonal water cycle in the MC plays a major role in creating the boreal-winter-spring freshwater plug, with the main source for the freshwater plug contributed directly by precipitation over the Java Sea and runoff from Kalimantan.

**Upper-layer salinity budget**. To quantify the effect of Java-Sea precipitation and the inferred role of runoff from Kalimantan on the seasonal freshwater plug, a budget analysis was performed for the seasonal variations of upper-layer salinity in the Java Sea (see Upper-layer salinity budget in the Methods section). A rectangular box encompassing the middle of the Java Sea (yellow box in Fig. 3) is selected for the salinity budget analysis. This region provides an important source of seasonal freshwater anomaly that is advected into Makassar Strait by ocean currents. We cannot select a box that extends to the coasts of Kalimantan in the north and Java in the south because the SMAP SSS within 40km of the coasts are masked out due to potential contamination by land signals (see Methods).

The time series of the budget terms (Fig. 7a) show the variations of the salinity tendency and the contributions by

precipitation, horizontal advection, and a residual term that includes the effects of oceanic vertical processes (e.g., vertical mixing), evaporation, and the errors of the estimations for the other three terms that are estimated explicitly. Although the differences among the 3 years are noticeable for various budget terms, the dominance of the seasonal variability is evident. The slightly more positive values of precipitation tendency (blue curve) and SSS tendency (black curve) in 2015 than in 2017 indicate the effects of the IOD and El Nino that tend to reduce precipitation over the Maritime Continent region, which causes less SSS freshening. However, these changes are much smaller than the magnitude of the seasonal variability. We therefore produce averaged seasonal cycle for each term by averaging the values in the same month of different years from 2015 to 2018 (Fig. 7b). The seasonal variations were plotted twice to facilitate the visualization.

The effects of -P and horizontal advection generally assist the seasonal variation of SSS tendency while the residual term generally counteracts the combined effects of -P and horizontal advection, especially during DJF and MAM. The counteracting effect of the residual can be understood as the dissipative influence of vertical mixing (and evaporation to a lesser extent) that attenuate the upper-layer freshening. The time-averaged effect of horizontal advection is nearly zero because the currents are dominated by seasonal variations.

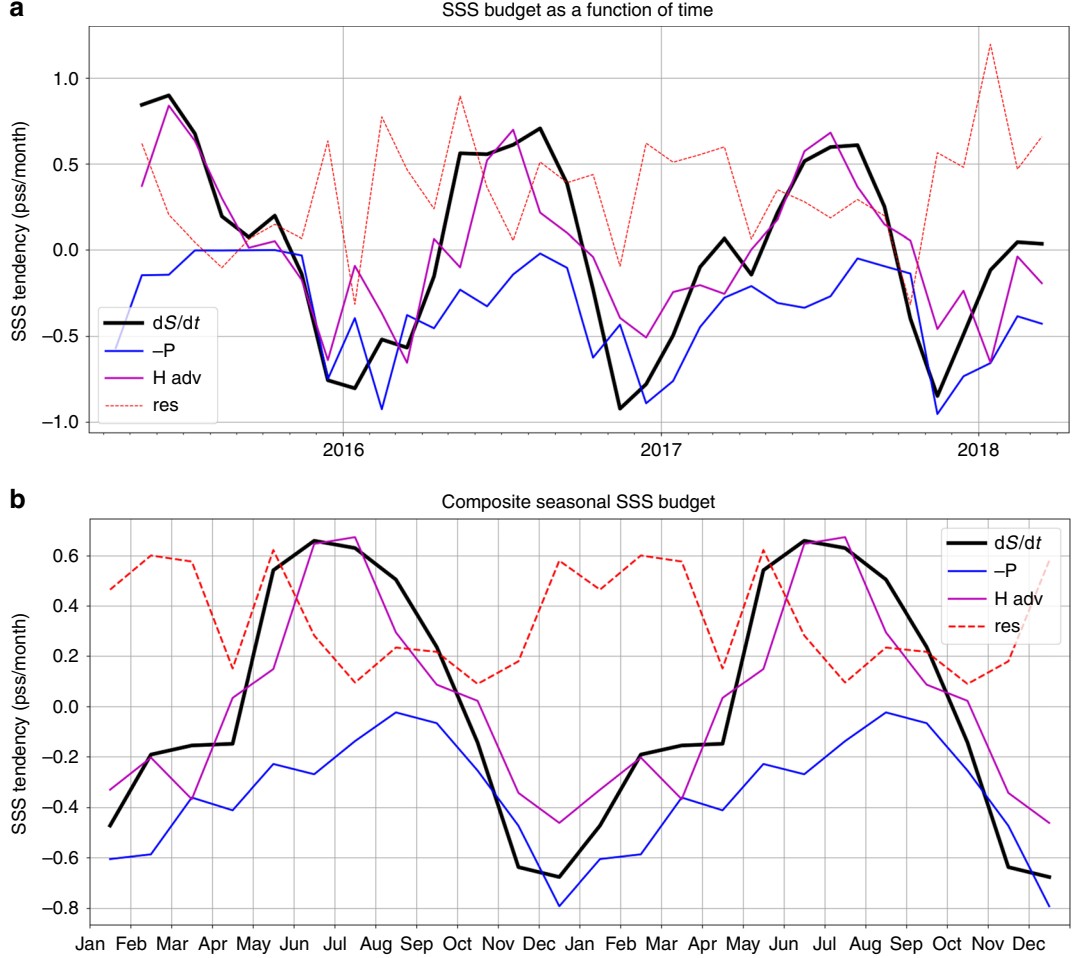

**Fig. 7** Time series of sea surface salinity budgets for the Java Sea and the averaged seasonal cycle. **a** Budget terms for the 3-year period of the study. **b** Seasonal cycles by averaging the values of the same month from different years. The seasonal cycles were plotted twice to facilitate the visualization. Black curves indicate the observed sea surface salinity tendencies. The blue, magenta, and red dashed curves represent the contributions by precipitation, horizontal advection, and the residual, respectively

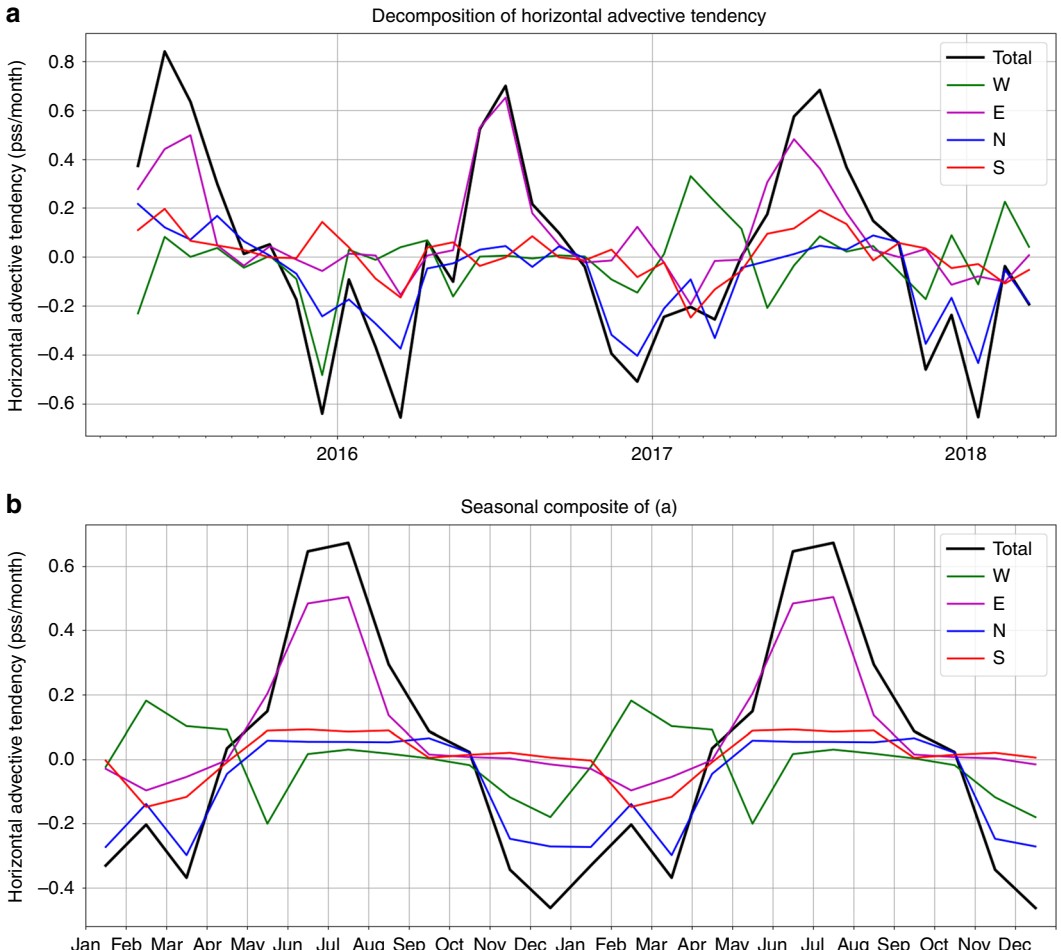

**Fig. 8** Decomposition of horizontal advective tendency. The total (black) horizontal advective tendency is divided into the contributions through the Western (green), Eastern (purple), Northern (blue), and Southern (red) interfaces. **a** Time series for the 3-year period of the study. **b** Seasonal cycles by averaging the values of the same month from different years. The seasonal cycles were plotted twice to facilitate the visualization

From DJF and MAM, the effect of -P (blue curve) results in significantly more negative SSS tendencies (freshening) than the observed SSS tendencies (black curve). This is further re-enforced by the smaller freshening effect due to horizontal advection (magenta curve). The combined effect of -P and horizontal advection is partially compensated by the residual term to give rise to the observed SSS tendencies during these seasons. During JJA (the dry season for the Indonesian Seas), -P has little contribution whereas the effect of horizontal advection explains most of the positive SSS tendencies (salinification).

To understand the effect of horizontal advection on SSS tendencies as shown in Fig. 7, we present the contributions from the advection through the four interfaces (Fig. 8) (see Upper-layer salinity budget analysis in the Method section). Similar to Fig. 7, the interannual time series are shown in the upper panel while the averaged seasonal variations are illustrated in the lower panel. During DJF and MAM, the total horizontal advection is primarily contributed by the advection across the northern interface. This is because the southward flow into the box through the northern interface (Fig. 3a, b) brings in waters from the northern interface (coming from the coastal oceans south of Kalimantan) that have lower salinity than the averaged salinity inside the box (e.g., Fig. 2a, b), thereby creating a freshening tendency. This advective freshening tendency from the northern interface reflects the influence of the runoff from Kalimantan that re-enforces the freshening in the Java Sea during DJF and MAM.

Because the runoff from Kalimantan is caused by the precipitation over the island during DJF and MAM (Fig. 4a, b), the precipitation associated with the monsoonal water cycle over the Java Sea and Kalimantan together is the major cause of the seasonal freshening in the Java Sea. The Java Sea freshening spreads into the southern Makassar Strait (Fig. 2a, b), which is further re-enforced by the runoff from the southeast coast of Kalimantan (also evident from the CDOM ocean color data shown in Fig. 6b). Therefore, we conclude that the MC monsoonal water cycle is the primary cause of the seasonal freshwater plug. Although the SCS throughflow brings waters into the Java Sea that are fresher than those carried by the inflow from the northwest Pacific, they are insufficient to explain the strong freshening observed in the Java Sea. However, the monsoonal wind-driven currents associated with the SCS throughflow provide a dynamical process to spread the Java Sea freshening into the Makassar Strait.

For completeness, we also investigate the nature of horizontal advection during JJA (dry season over the Java Sea sector). During this season, the major contribution to the total horizontal advection is due to that across the eastern interface (Fig. 8b). This is because the westward flow into the box through the eastern interface (Fig. 3c) brings waters from the eastern interface that have higher salinity than the average salinity inside the box (Fig. 2c), thereby causing a positive salinity tendency.

To our knowledge, the only study that investigated the seasonal budget of upper-ocean salinity in the Indonesian Seas is a model-based study[48] that analyzed the upper 50-m salinity budget in various basins in the SEAS. The finding from that study is that seasonal variation of the salinity in the upper 50 m was generally caused by a combination of surface freshwater flux and three-dimensional velocities. Consistent with that study, our observational analysis also shows the importance of precipitation and ocean dynamics in regulating the seasonal SSS in the central Java Sea. Moreover, we identified precipitation over the Java Sea and runoff from Kalimantan as the dominant sources for the seasonal freshwater plug, and characterized the nature of the contributions by horizontal advection.

**Downstream effects on the Makassar Strait.** The seasonal freshwater plug affects the dynamic height and results in an upstream pressure gradient along the Makassar Strait in the upper layer, thereby inhibiting the southward flow in the upper layer of the ITF[17]. This effect has not been documented using observations across the entire span of the Makassar Strait. Here we illustrate this effect using altimetry-derived sea level anomaly (SLA) (see Data in the Methods section) and SMAP SSS measurements. The seasonal composites of SLA (Fig. 9) show that the SLA decreases from the eastern Java Sea south of the Makassar Strait to the western Sulawesi Sea north of the Makassar Strait during DJF and MAM (Fig. 9a, b). This tendency is reversed during JJA and SON (Fig. 9c, d).

To quantify the meridional variation of SLA and SSS across the Makassar Strait, we selected the rectangular box shown in Fig. 9 that straddles across the Makassar Strait, and computed the meridional profiles of SLA and SSS zonally averaged within that box. The resultant meridional SLA and SSS profiles (Fig. 10a, b respectively) show that the SLA decreases by approximately 10.5 cm from 4 °S to 4 °N. during DJF. The sharpest rate of decrease occurs near the southern Makassar Strait: an approximately 4.5-cm drop from 4 °S to 2 °S. During MAM, there is a similar drop of SLA from 4 °S to 2 °S. The SSS generally increases northward during DJF and MAM (Fig. 10b). This is overall consistent with the northward decrease of SLA, especially between 4 °S and 2 °S because fresher water causes a larger dynamic height in the upper layer.

At the 4 °S–2 °S latitude range, the decrease of SLA is similar between DJF and MAM (4–5 cm) (Fig. 10a). However, the increase of SSS is more than three times as large in MAM than that in DJF (Fig. 10b). The larger northward increase of SSS in MAM is not associated with a larger decrease of SLA because of two possible factors: (1) thermosteric contribution to SLA compensates some halosteric contribution during MAM, and (2) difference in the density structure of the deeper ITF flow between DJF and MAM. To test (1), we show the meridional profiles of sea surface temperature (SST) (Fig. 10c) for the same box. During MAM, there is a northward decrease of SST. This would have caused a northward decrease of SLA (due to a reduction in thermal expansion) that re-enforces the halosteric contribution to SLA. But this expected outcome contradicts the SLA data that show a similar SLA decrease between MAM and DJF. Therefore, we conclude that deeper density changes of the ITF must have compensated part of the upper-layer halosteric and thermosteric contributions to SLA. This does not mean that the freshening during MAM in the southern Makassar Strait has less influence on the upper ITF than the freshening during DJF. Rather, it is an indication of a compensating effect due a to density contribution below the surface layer. Without the freshening in MAM, the ITF vertical structure would have been different. In fact, velocity observations from a mooring in the Makassar Strait show that upper-layer southward flow of the ITF rebound very slowly during MAM after the boreal-inter

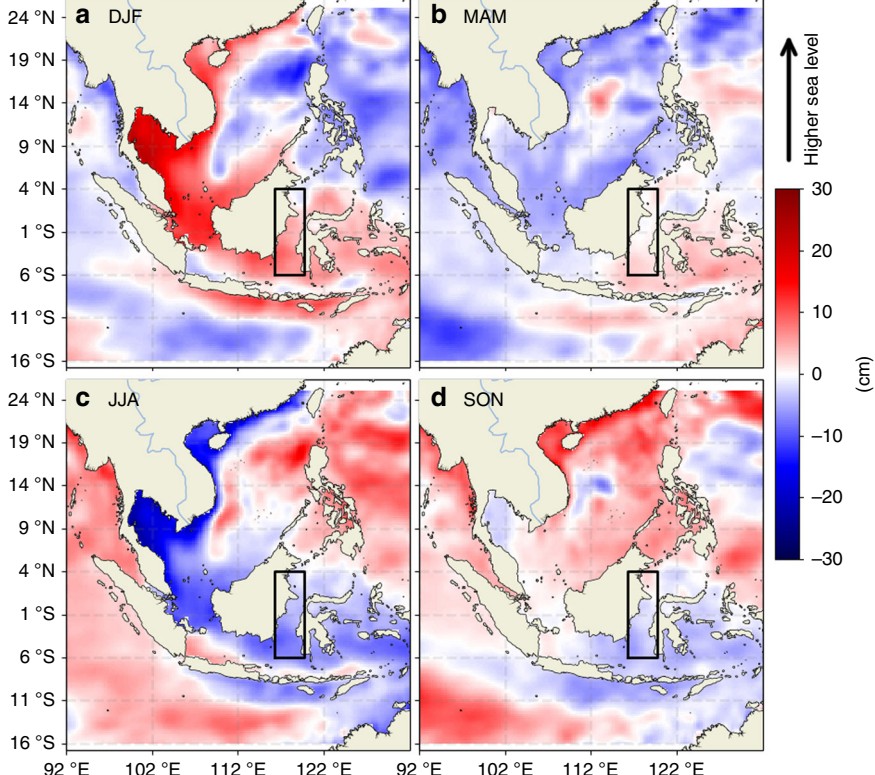

**Fig. 9** Composite seasonal maps of sea level anomalies. **a–d** correspond to the maps for the DJF, MAM, JJA, and SON seasons. The rectangular box (116.25–119.75 °E, 6S-4 °N) indicates the region where the meridional profiles are shown in Fig. 10

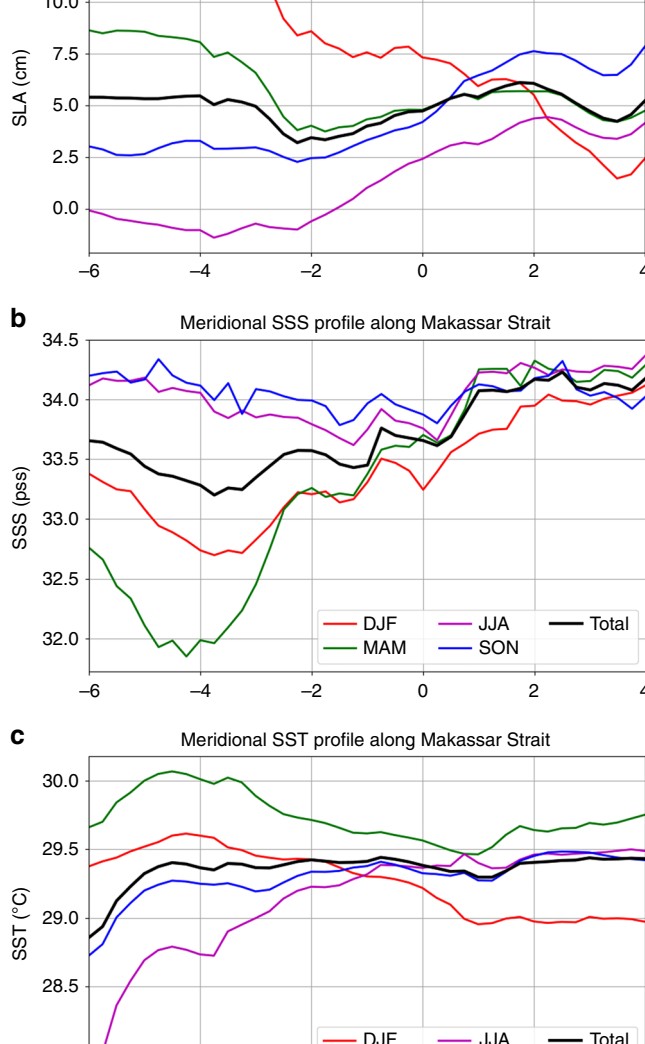

**Fig. 10** Meridional profiles of sea level anomalies, sea surface salinity, and sea surface temperature along the Makassar Strait. **a** Sea level anomalies. **b** Sea surface salinity. **c** Sea surface temperature. The data shown correspond to the zonally averaged across the box shown in Fig. 9

weakening caused by the DJF freshening in the upper layer[49]. This slow rebound may be contributed by the additional freshwaters in MAM due to runoff from Kalimantan.

To examine the extent to which SSS and SST variations affect the meridional SLA gradient, we estimated the contributions of SSS and SST to steric height in a mixed layer with a depth of 50 or 25 m (see SSS and SST contributions to SLA gradient along the Makassar Strait in the Methods section). The results suggest that the boreal-winter-spring freshwater plug is more than enough to explain the observed enhancement of south-to-north (upstream) SLA gradient. SST variations are found to have little contributions to steric height. For a 25-m mixed-layer depth, the freshwater plug would still explain more than 50% of the enhanced south-to-

north SLA gradient during the freshwater-plug seasons. Because the seafloor depth varies dramatically across the Makassar Strait with the bulk of the deeper ITF flowing primarily in the eastern part[50,51], we also conducted similar calculations for the eastern and western halves of the Strait separately. The results are very similar to those for the entire Strait.

## Discussion

This study reveals the critical role of the regional water cycle of the Maritime Continent in regulating the Indonesian through-flow, an ocean current linking the Indo-Pacific that has been widely recognized to be important to global ocean circulation, climate, and biogeochemistry. Specifically, we use a suite of satellite observations of the ocean, atmosphere, and land along with ocean surface current estimates to characterize the spatial and temporal evolution of the seasonal freshwater plug in the Indonesian Seas, identify the sources of the seasonal freshwater plug and the relationship with the regional monsoonal water cycle in the MC, and investigate the relationship of the seasonal freshwater plug with the meridional pressure gradient in the Makassar Strait that regulates the ITF. The results suggest that the freshwater plug not only occurs in boreal winter (DJF), but persists into boreal spring (MAM). The freshwater input due to boreal-winter monsoonal precipitation over the Java Sea is found to be a major source of the freshwater plug. Runoff from Kalimantan caused by the boreal-winter–spring precipitation over the island helps re-enforce the freshwater plug and prolong it into boreal spring. The boreal-winter–spring freshening is associated with an upstream decrease of sea level anomaly along the Makassar Strait, reflecting the modulation of the along-strait pressure gradient that influences the ITF. Put together, the results suggest that the MC monsoonal water cycle plays a critical role in regulating the low-latitude chokepoint of global ocean circulation through its effect on the freshwater plug the subsequent impact on the ITF. Our finding is in contrast to the previous suggestion that SCS waters is the source of the boreal-winter freshwater plug. The freshening effect due to the SCS waters is insufficient to explain the large freshening observed in the Java Sea. The observations presented here and the underlying identified physical processes that control the seasonal freshwater plug are useful for evaluating ocean and climate models and the projected changes using the latter. Were the mechanism for the linkage of the ocean circulation and water cycle not correctly represented by climate models, the fidelity of the projected changes based on such models would be questionable.

Our study also motivates further investigations of the relative contribution of different sources for the freshwater and the related impacts on the ITF on longer time scales. On interannual time scales, while the SCS waters entering the Makassar Strait through the Sulawesi Sea associated with El Niño events can modify the ITF[18], the precipitation over the Sulawesi Sea associated with El Niño may also influence the upper-ocean salinity in the Sulawesi Sea. On decadal time scales, the strengthening of the ITF in the past decade has been associated with an enhanced precipitation in the MC region[52]. The associated details of the water cycle processes (e.g., sources of regional precipitation and runoff) warrants further investigation. Longer-term climate variability and change in the Indo-Pacific sector, e.g., associated with changes in the Walker Circulation and the effects on atmospheric convection in the MC region[53,54], could modify the MC water cycle and influence the freshwater effects on the ITF. To better understand the longer-term variability and linkage between the ocean and water cycle, it is critical to sustain satellite SSS such as those from SMAP that have provided an

unprecedented capability to resolve SSS variability in the MC, including the SSS in various straits.

## Methods

**Data**. Our analysis is primarily based on satellite measurements of the ocean, atmosphere, and land. These include measurements of SSS, precipitation, soil moisture, ocean color, sea level anomaly (SLA), and sea surface temperature (SST). In addition, we use ocean surface current estimates from a high-resolution global ocean analysis, evaporation estimates based on a synthesis product of satellite and atmospheric reanalysis datasets, and an in-situ-based SSS climatology.

The satellite SSS are used to characterize the spatial structure and seasonal evolution of the SSS. The precipitation (P) and evaporation (E) products are used to assess the relative contribution of E and P to the evaporation−precipitation (E−P) and their impacts on SSS variability. The ocean surface currents together with SSS data are used to estimate the horizontal advective tendencies for SSS. Because river runoff from the MC islands are not publicly available, we use precipitation over land, soil moisture, and ocean color to elucidate the effect of river runoff on SSS. The SSS and SLA measurements are used to illustrate the relationship of SSS variability with that of the meridional pressure gradient along the Makassar Strait, which exemplifies the impact on the ITF. The main attributes of various datasets are described below.

The SMAP SSS product used in this study is the level-3, version-4, 8-day running mean SMAP SSS retrieval distributed by the Jet Propulsion Laboratory[55]. It is distributed by NASA Physical Oceanography Distributed Active Archive Center (PO.DAAC): (https://podaac.jpl.nasa.gov/dataset/SMAP_JPL_L3_SSS_CAP_8DAY-RUNNINGMEAN_V4). This 40-km resolution product, combining measurements from the ascending and descending orbits, is mapped on a 0.25° grid. The data in the first 40 km from the coast were masked out due to the potential contamination of SSS retrieval by land signals. We also use soil moisture retrieval from SMAP, which is the radiometer-based, level-3, 0.25°-grid, daily product[56] obtained from the National Snow and Ice Data Center (NSIDC, nsidc.org): (ftp://n5eil01u.ecs.nsidc.org/SAN/SMAP/SPL3SMP.004/).

As a consistency check, we have also examined the level-3, version-3, de-biased SMOS SSS product generated by the Laboratoire d'Océanographie et du Climat (LOCEAN)[28] and distributed by the Centre National d'Etudes Spatiales (CNES) and Institut Français de Recherche pour l'Exploitation de la Mer (IFREMER) Centre Aval de Traitement des Données SMOS (CATDS). The product is available through https://www.catds.fr/Products/Available-products-from-CEC-OS/CEC-Locean-L3-Debiased-v3. The SMOS SSS product has a longer record (2010–2017) than the SMAP SSS product (April 2015-present). The seasonal variability of SMOS SSS is found to be overall consistent with that from SMAP SSS, regardless whether we use the entire SMOS SSS record or the SMOS SSS during the SMAP period. However, SMOS SSS are noisier than SMAP SSS in the MC region (not shown). Therefore, the quantitative results presented in this study are based on the SMAP SSS for the period of April 2015–March 2018. Despite the shortness of the SMAP SSS record, we believe that the finding about the seasonal SSS variability to be robust based on the overall similarity of the shorter SMAP and longer SMOS SSS.

The monthly climatology of 0-m salinity from the World Ocean Atlas 2013 version 2 (WOA13) is used for a comparison with the satellite SSS. The dataset is available through (https://www.nodc.noaa.gov/cgi-bin/OC5/woa13/woa13.pl?parameter=s). This climatology is based on measurements from the 1955–2012 period mapped to a 1° × 1° grid[57]. The first two standard depths of WOA13 are 0 and 5 m. Presumably, the measurements above 2.5 m were used to generate the 0-m salinity. For the sake of convenience, we will refer to the 0-m salinity from WOA13 as the SSS. However, we are mindful about the potential near-surface salinity stratification that can cause differences between the SSS measured by satellites in the upper 1–2 cm and the in situ near-surface salinity that is more typically measured at depths deeper than 1 m. Such differences are on the order of 0.1 pss on time scales longer than a few days[58]. This is more than an order of magnitude smaller than the amplitude of seasonal variation of SSS in the MC region as presented in this study. Moreover, strong tidal mixing occurs in the Indonesian Seas[9], including the very shallow regions such as the Karimata Strait and the Java Sea. Therefore, the near-surface salinity stratification is expected to be small.

The precipitation product used is the version-7 Tropical Rainfall Measuring Mission (TRMM) Multi-satellite Precipitation Analysis (TMPA) 3B42 product generated and distributed by Goddard Space Flight Center (GSFC) Distributed Active Archive Center (https://pmm.nasa.gov/data-access/downloads/trmm). The product combines precipitation measurements from TRMM and those from the Global Precipitation Measurement (GPM) mission core satellite to produce a two-decade record. We used the level-3, 0.25°-grid, daily accumulated precipitation dataset.

The SLA dataset is the level-3 product based on measurements from multiple satellite altimeters. The product was mapped at a daily interval onto a 0.25° grid. The SLA data are distributed by the Copernicus Marine Environment Monitoring Services: (http://marine.copernicus.eu/services-portfolio/access-to-products/?option=com_csw&view=details&product_id=SEALEVEL_GLO_PHY_L4_REP_OBSERVATIONS_008_047).

For evaporation, we use the estimates from the Objectively Analyzed Air-Sea Fluxes (OAFLUX) project (http://oaflux.whoi.edu)[59]. The OAFLUX evaporation is derived from satellite measurements of wind and SST as well as surface air temperature and humidity from atmospheric reanalysis. The product is monthly on a 1° grid.

The SST product we used is the level-4 daily Operational Sea Surface Temperature and Sea Ice Analysis (OSTIA) SST analysis produced by the United Kingdom Met Office on a global 0.054° grid. The product is available through the PO.DAAC: https://doi.org/10.5067/GHOST-4FK02.

For ocean color, we used the Moderate Resolution Imaging Spectroradiometer (MODIS) level-3, daily, 9-km resolution absorption coefficient of colored dissolved and detrital organic matter (CDOM) at 443 nm ($a_{cdm}$) data (2002−present), processed and distributed by the NASA GSFC (https://oceancolor.gsfc.nasa.gov/). This $a_{cdm}$ product is generated using the Quasi Analytical Algorithm (QAA)[60].

The ocean surface current estimates from the Hybrid Coordinate Community Ocean Model (HYCOM) 1/12° global ocean analysis are also used in this study. The product is available through https://www.ncdc.noaa.gov/data-access/model-data/model-datasets/navoceano-hycom-glb. The HYCOM analysis (2013–present) assimilates SLA observations from altimeters, satellite, and in situ sea SST, in situ vertical profiles of temperature and salinity from Argo floats and moored buoys, and temperature profiles from eXpendable BathyThermograph (XBT). However, there is a general lack of in situ salinity data in the MC region. The assimilation was performed using the US Navy Coupled Ocean Data Assimilation (NCODA) system. The HYCOM currents were used because they represent the currents in the MC region relatively well[51]. We used the 0–20 m averaged HYCOM currents for our analysis (referred to as HYCOM surface currents in this paper). However, the results based on the top-layer currents (at 1 m) produced similar results.

Although all other datasets used in this study have a longer record than that of SMAP, our analysis focuses on 3 complete years covered by SMAP from April 2015 to March 2018. All datasets encompass this period, except for the OAFLUX evaporation that is currently available up to only December 2017. We found that the seasonal variation of E−P in the MC region is heavily dominated by P, with the seasonal variation of -P larger than that of E by almost an order of magnitude in the Java Sea. Therefore, our discussion focuses on the effect of P.

The bathymetry data used in Fig. 1 were obtained from https://www.ngdc.noaa.gov/mgg/global/etopo2.html.

**Upper-layer salinity budget**. The upper-layer salinity budget for the box in the Java Sea (Fig. 3) can be expressed by the following equation:

$$\frac{H}{\text{Vol}} \times \iint \frac{dS}{dt}\,dxdy = \frac{S_0}{\text{Vol}} \iint (-P)\,dxdy + \text{Hadv} + \text{RES}, \qquad (1)$$

where $H$ is assumed to be 20 m, the approximate mixed-layer depth during DJF and MAM in the Java Sea[61], Vol is volume of the box, $S_0$ is mean salinity (35 pss) that converts surface freshwater flux to salinity tendency unit. The left-hand side is the averaged SSS tendency inside the box. The first term on the right-hand side is the effect of precipitation on SSS tendency. Hadv denotes the effect of horizontal advection on the volume-averaged salinity tendency. RES is the residual that contains other effects such as those due to vertical processes, evaporation (E), and the errors associated with the estimations of the other three terms. We did not include the effect of E in the first term on the right-hand side because (1) the evaporation product from OAFLUX does not cover the entire 3-year period of the study when this manuscript is written and (2) seasonal variation of E is smaller than that of -P by almost an order of magnitude in the Java Sea.

The total horizontal advective tendency through the interfaces of the box is given by the following equation based on the method of characterizing heat advection through various interfaces of a box[62]:

$$\text{Hadv} = \frac{H}{\text{Vol}} \Big[ \int u_W(S_W - S_{\text{Ref}})dy - \int u_E(S_E - S_{\text{Ref}})dy \\ + \int v_S(S_S - S_{\text{Ref}})dx - \int v_N(S_N - S_{\text{Ref}})dx \Big], \qquad (2)$$

where the four terms denote the contributions from the western, eastern, southern, and northern interface respectively, with the subscript W, E, S, and N indicating the four interfaces. All quantities except for the depth $H$ and volume Vol are treated as a function of time. $S_{\text{Ref}}$ is the box-averaged SSS at any particular time. The physical meaning of the terms in Eq. (2) is intuitive: if the velocity at an interface brings in waters that has higher (lower) salinity than the averaged salinity of the box ($S_{\text{Ref}}$) at a given time, it would increase (decrease) the averaged salinity of the box, i.e., causing a positive (negative) tendency for the box-averaged salinity. Therefore, the terms in Eq. (2) describe the relative impacts of horizontal advection through various lateral interfaces on the time-dependent, box-averaged salinity tendency. Readers are referred to ref. [62] for the theoretical derivation and how it relates to the box average of the traditional form of local (point-wise) advective tendencies. That study pointed out that the traditional form of local advective tendencies averaged over a box only describes the averaged advective redistribution of heat (or salt) within the box due to advective tendencies inside the box. If there is smaller-scale variability within the box, the traditional form of box-averaged local advective tendencies cannot characterize the advection of heat (or salt) through various interfaces that modify the average temperature (or salinity) of the box.

However, the formulation developed by ref. [62] explicitly quantifies the advective sources of heat (or salt) through the various interfaces of the box.

**SSS and SST effects on SLA gradient in the Makassar Strait**. To assess the relative contributions of meridional changes in SSS and SST to the meridional change in SLA along the Makassar Strait on seasonal time scales, we estimated the contributions of SSS and SST anomalies shown in Fig. 10b, c to steric height using the following equations:

$$\eta(S)' = -\beta S'H, \qquad (3)$$

$$\eta(T)' = \alpha T'H, \qquad (4)$$

where $S'$ is SSS anomaly, $T'$ is SST anomaly, $\beta$ is the saline compression coefficient, $\alpha$ is the thermal expansion coefficient, $H$ is the mixed-layer depth, $\eta(S)'$ denotes the contribution of $S'$ in the mixed layer to dynamic height (halosteric height), and $\eta(T)'$ represents the contribution of $T'$ in the mixed layer to dynamic height (thermosteric height). The values of $\beta$ and $\alpha$ are $7.36 \times 10^{-4}$ psu$^{-1}$ and $3.11 \times 10^{-4}$ C$^{-1}$, respectively[63]. We assume $H = 50$ m in the lack of in situ measurement.

From Fig. 10b, c, we calculated the meridional difference of $S'$ and $T'$ between the southern end of the Makassar Strait (4 °S) and northern end of the Strait (2 °N), averaged over the freshwater-plug seasons (DJF-MAM). These meridional differences of $S'$ and $T'$ are plugged into Eqs. (3) and (4) to obtain the meridional difference of $\eta(S)'$ and $\eta(T)'$ (denoted as $\Delta\eta(S)'$ and $\Delta\eta(T)'$) between the southern and northern ends of the Strait. The results are $\Delta\eta(S)' = 6.5$ cm and $\Delta\eta(T)' = 0.8$ cm. The observed meridional difference of SLA along the Strait (denoted as $\Delta$SLA) between 4 °S and 2 °N is 4.9 cm. Therefore, the steric height caused by the seasonal freshwater plug (6.5 cm) is more than enough to explain the 4.9-cm observed SLA gradient during the freshwater-plug seasons. SST has very little contribution to steric height. Even for a 25-m mixed-layer depth, the freshwater plug would still create 3.3 cm of meridional steric height gradient, which is more than 50% of the observed SLA gradient. We also performed similar assessment for the eastern and western halves of the Strait using the mid ocean point at each latitude to divide the Strait. The results are very similar to those for the entire Strait: (a) for the eastern half: $\Delta\eta(S)' = 6.7$ cm, $\Delta\eta(T)' = 0.8$ cm, observed $\Delta$SLA = 5.8 cm; (b) western half: $\Delta\eta(S)' = 5.1$ cm, $\Delta\eta(T)' = 0.7$ cm, observed $\Delta$SLA = 4.7 cm. Therefore, the seasonal freshwater plug being a major factor in regulating the meridional SLA gradient is a very robust result.

## Data availability
The authors declare that the data supporting the findings of this study are available within the article (through the hyperlinks for the respective datasets). Extra data are available from the corresponding author upon request.

## Code availability
All computer codes for the analysis of the data are available to readers upon request without restriction.

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

## Acknowledgements

The research described in this paper was carried out in part at the Jet Propulsion Laboratory, California Institute of Technology, under a contract with NASA. T.L. and S. F. are supported by NASA Physical Oceanography Program. A.L.G. acknowledges the grant N00014-16-1-2480 "Low Latitude Indian to Western Pacific, 40° to 160° E" from the Office of Naval Research. J.S. acknowledges support from the National Science Foundation under Grant Number OCE-1736285. Lamont-Doherty contribution number 8305.

## Author contributions

T.L. conceived the motivation and objectives. T.L. and S.F. planned the analyses. S.F. performed the analyses. A.L.G. and J.S. provided expert knowledge of maritime continent circulation and guidance on the discussion in relation to existing literature. All authors contributed to conceiving figures. S.F. produced figures. T.L. wrote the initial draft of the paper and all authors contributed to subsequent revisions.

## Additional information

**Competing interests:** The authors declare no competing interests.

