## [Peer Review File · Nature Communications]

Reviewers' comments:

Reviewer #1 (Remarks to the Author):

Review of the manuscript entitled "Maritime continent water cycle regulates low-latitude chokepoint of global ocean circulation" by T. Lee et al.

The study by Lee et al investigates the nature and origin of the so-called 'freshwater plug' into Makassar Strait, a primary pathway for the Indonesian Throughflow (ITF). The 'plug', observed in boreal winter and spring, results in reduced along-strait pressure gradient and weakened transport within the upper layer of ITF. The authors demonstrate that the major source of freshwater for the 'plug' is local; that is, it is due to surface freshening in the Java Sea due to local precipitation and runoff from Kalimantan, Indonesia. The results emphasize a critical role played by the local hydrological cycle in regulating the seasonality of ITF. This is also in construct to the previous studies which pointed to the South China Sea waters as the source of the seasonal 'freshwater plug'. The findings have important implications for understanding the physical mechanisms responsible for regulating ITF and the role of ITF in the Indo-Pacific climate variability.

The paper is well-written, logically organized, and the results are novel and significant. The study is based mostly on observational data, utilized in a clever way, which makes it particularly appealing. I think this study will be of significant interest to a broad scientific audience and I recommend publication. Although the paper can go 'as is', in my view, there are a few minor comments the authors might want to address.

Figure 1 is hard to read. Maybe it is just my eyes, but I find the use of the same colors to show topography (very important figure in this paper) and SSS anomalies (Fig. 2) a bit confusing. Very minor, but I personally like how geography is presented in e.g., Fig 1 in Ref [18].

Please explain why model (HYCOM) velocities are used and how this may affect the conclusions. The use of model velocities is a bit in contrast to the intro statement "Moreover, most ocean models offer little prospect for elucidating the mechanism controlling SSS..." The model fails to reproduce SSS structure and variability, which, in turn, is important in controlling flows through the straits (the effect of 'freshwater plug'), yet, somehow, the model is able to produce 'true' velocities through the same straits.

Just noticed:

Line 90: delete "are"

Line 216: "are" instead of "is"

Line 330: "which" or "that" after "...Kalimantan)"

Line 553: the product is on a 1-deg grid

Reviewer #2 (Remarks to the Author):

This paper investigated the contributions of the MC monsoonal water cycle to the boreal winter-spring freshening in the Java Sea based on a salinity budget analysis. The results show that the precipitation and ocean dynamics play important roles in the seasonal freshwater plug in the Java Sea. Furthermore, this paper documented the relationship between the seasonal freshwater plug

and the meridional pressure gradient in the Makassar Strait that regulates the ITF. These findings are interesting and important that would expand our understanding of the impacts of the MC water cycle on the variations of the ITF on seasonal to longer-term time scale. The paper is overall well written and presented. I have some comments below for consideration to make the results more convincing.

1. Since only 3 years data are used, the comparison of mean status is important. I suggest to include the 3 years means and their long term climatology mean in all seasonal figures.
2. The spatial and temporal of the seasonal freshwater plug in the Indonesian Seas are well captured by the satellite observations, despite the shortness of the data record. I am wondering whether the results of salinity budget in the Java Sea on seasonal time scale are affected by the interannual signals (IOD or ENSO, especially the 2015-2016 strong El Nino event) ? Even though the differences in salinity budget terms on seasonal and interannual time scales seem to be small.
3. I suggest to conduct a statistical significance test to make the results of the salinity budget more convincing, or try one more product, e.g. XBT data.
4. Is there a consistency between the seasonal variation of the SSS south of the Makassar Strait and that of the Java Sea? Is only the SSS in the Java Sea representative in the boreal winter and spring?
5. Could the analysis of the ocean dynamic process be simplified, which would help us better understand the impacts of the ocean dynamic process, if the coasts of the Java and Kalimantan Islands are used as the north and south interface of the control volume of the salinity budget?
6. As Figure 10 shown, the low-salinity core appears around 4°S. Is there a relationship with CSC current?
7. I understand the difficulties of quantifying the halosteric and thermosteric contribution to SLA. How much influence does the change in meridional pressure gradient have on ITF? Can it be quantified?

Reviewer #3 (Remarks to the Author):

This study demonstrates the utility of salinity remote sensing data in the Indonesian Seas to understand the interaction between seasonal water cycle and the Indonesian Throughflow. In contrast to previous studies suggesting monsoonal freshwater input from the South China Sea through Karimata Strait, this study shows that the boreal winter-spring freshening of surface water parcels in the Makassar Strait is caused by local precipitation and run-off from Kalimantan, which leads to a reduced southward pressure gradient that would weaken the Indonesian Throughflow. I think this study meets a major challenge in oceanography for the Indonesian Seas by carefully using latest remote sensing datasets and thus recommend publication as is.

Response letter for manuscript NCOMMS-18-26394A
(Reviewers' comments in black, response in blue)

Reviewer #1 (Remarks to the Author):

Review of the manuscript entitled “Maritime continent water cycle regulates low-latitude chokepoint of global ocean circulation” by T. Lee et al.

The study by Lee et al investigates the nature and origin of the so-called ‘freshwater plug’ into Makassar Strait, a primary pathway for the Indonesian Throughflow (ITF). The ‘plug’, observed in boreal winter and spring, results in reduced along-strait pressure gradient and weakened transport within the upper layer of ITF. The authors demonstrate that the major source of freshwater for the ‘plug’ is local; that is, it is due to surface freshening in the Java Sea due to local precipitation and runoff from Kalimantan, Indonesia. The results emphasize a critical role played by the local hydrological cycle in regulating the seasonality of ITF. This is also in construct to the previous studies which pointed to the South China Sea waters as the source of the seasonal ‘freshwater plug’. The findings have important implications for understanding the physical mechanisms responsible for regulating ITF and the role of ITF in the Indo-Pacific climate variability.

The paper is well-written, logically organized, and the results are novel and significant. The study is based mostly on observational data, utilized in a clever way, which makes it particularly appealing. I think this study will be of significant interest to a broad scientific audience and I recommend publication. Although the paper can go ‘as is’, in my view, there are a few minor comments the authors might want to address.

We greatly appreciate the positive and constructive comments from the reviewer.

Figure 1 is hard to read. Maybe it is just my eyes, but I find the use of the same colors to show topography (very important figure in this paper) and SSS anomalies (Fig. 2) a bit confusing. Very minor, but I personally like how geography is presented in e.g., Fig 1 in Ref [18].

We revised Figure 1 to use a color scheme similar to Figure 1 in Ref[18] as suggested, which makes the figure not as busy and the colors of the bathymetry more intuitive. We left Figure 2 as is because the color scheme is common to many published studies for salinity.

Please explain why model (HYCOM) velocities are used and how this may affect the conclusions. The use of model velocities is a bit in contrast to the intro statement “Moreover, most ocean models offer little prospect for elucidating the mechanism controlling SSS...” The model fails to reproduce SSS structure and variability, which, in turn, is important in controlling flows through the straits (the effect of ‘freshwater plug’), yet, somehow, the model is able to produce ‘true’ velocities through the same straits.

As we discussed in the Method section, the HYCOM data assimilation product represented the currents in the maritime continent region fairly well (Ref 51). In fact, HYCOM SSS also resembles the particular SSS climatology towards which HYCOM relaxes its model SSS.

However, the relaxation term is a statistical correction of the model SSS, thereby not representing any specific physical processes. Therefore, it precludes an analysis of the physical mechanism for the SSS budget. To clarify this, we revised the sentence originally in lines 103-104 as “The SSS relaxation, which is a statistical correction to compensate the errors due to surface forcing and model deficiencies, complicates the investigation of the relationship between SSS and precipitation using model output.” (new lines 107-109 in the revised manuscript).

Just noticed:

Line 90: delete “are”

Fixed as suggested.

Line 216: “are” instead of “is”

Fixed as suggested.

Line 330: “which” or “that” after “...Kalimantan)”

Fixed as suggested.

Line 553: the product is on a 1-deg grid

Fixed as suggested.

Reviewer #2 (Remarks to the Author):

This paper investigated the contributions of the MC monsoonal water cycle to the boreal winter-spring freshening in the Java Sea based on a salinity budget analysis. The results show that the precipitation and ocean dynamics play important roles in the seasonal freshwater plug in the Java Sea. Furthermore, this paper documented the relationship between the seasonal freshwater plug and the meridional pressure gradient in the Makassar Strait that regulates the ITF. These findings are interesting and important that would expand our understanding of the impacts of the MC water cycle on the variations of the ITF on seasonal to longer-term time scale. The paper is overall well written and presented. I have some comments below for consideration to make the results more convincing.

We greatly appreciate the positive and constructive comments from the reviewer.

1. Since only 3 years data are used, the comparison of mean status is important. I suggest to include the 3 years means and their long term climatology mean in all seasonal figures.

Including the 3-year mean and long-term climatology (some parameters don't have long-term climatology) in all seasonal figures, essentially Figures 2-10, will significantly dilute the focus of the discussion on the boreal winter-spring freshwater plug, and make the graphical presentation and discussion extremely burdensome to the readers.

Moreover, as we discussed in the Method section, the seasonal variations of SMAP SSS for the 3-year period is similar to those of SMOS SSS over a 7-year period, suggesting that interannual variations of SSS are much smaller than those of seasonal variations (lines 527-529 in the revised manuscript). Also see our response to a related comment below that suggests the much smaller role of interannual variability than seasonal variability. We therefore respectfully decline this suggestion by the reviewer.

2. The spatial and temporal of the seasonal freshwater plug in the Indonesian Seas are well captured by the satellite observations, despite the shortness of the data record. I am wondering whether the results of salinity budget in the Java Sea on seasonal time scale are affected by the interannual signals (IOD or ENSO, especially the 2015-2016 strong El Nino event) ? Even though the differences in salinity budget terms on seasonal and interannual time scales seem to be small.

As we discussed in lines 303-304 of the revised manuscript, the time series of the budget terms for SSS anomalies for the three years are dominated by the seasonal cycle (Figure 7) despite the occurrence of the IOD and El Nino during 2015. Therefore, the seasonal budget for the boreal winter-spring freshwater plug inferred from the three-year averages is robust. We also added a couple of sentences pointing out that the SSS tendency and precipitation anomaly values in 2015 are only slightly more positive than those in 2017, reflecting the minor impacts of IOD and El Nino on the seasonal SSS budget (lines 304-308 in the revised manuscript). The secondary effects of the 2015 IOD and El Nino on precipitation and SSS over the greater southeast Asian Seas region will be reported in a follow-on paper by including the newly reprocessed SMOS SSS dataset that is 8 years long.

3. I suggest to conduct a statistical significance test to make the results of the salinity budget more convincing, or try one more product, e.g. XBT data.

As we pointed out in relation to comment#2, the seasonal budget terms are dominated by the seasonal cycle with very small year-to-year changes. We therefore believe the seasonal budget to be robust.

We cannot do salinity budget with XBT data because XBT only have temperature but not salinity measurements. If the reviewer is referring to XCDT measurements (that have both temperature and salinity), there were little (if any) such measurements in the Indonesian Seas, to our best knowledge. Shipboard CTD measurements in the region are extremely sparse and inhomogeneous as we pointed out in the Introduction region. In fact, the artifacts in the WOA13 climatology that we discussed in relation to Figure 2 reflect the impact of the inhomogeneity of the in-situ salinity measurements used to construct the climatology.

4. Is there a consistency between the seasonal variation of the SSS south of the Makassar Strait and that of the Java Sea? Is only the SSS in the Java Sea representative in the boreal winter and spring?

Figures 2a-b and Figures 3a-b showed that the SSS anomalies in the southern Makassar Strait are coherent with those in the Java Sea during boreal winter and spring. We also stated that “The Java Sea freshening spreads into the southern Makassar Strait (Figure 2a-b), which is further re-

enforced by the runoff from the southeast coast of Kalimantan” (lines 358-360 of the revised manuscript).

5. Could the analysis of the ocean dynamic process be simplified, which would help us better understand the impacts of the ocean dynamic process, if the coasts of the Java and Kalimantan Islands are used as the north and south interface of the control volume of the salinity budget?

We stated that “SMAP SSS data within one satellite footprint (40 km) of the coasts have been masked out to avoid the contamination of SSS data by land signals” (lines 254-256 of the revised manuscript). Due to the missing data near the coasts, we cannot select a control volume by using the coasts of the Java and Kalimantan Islands as the southern and northern boundaries. We added a sentence to further explain this (lines 296-298 in the revised manuscript).

6. As Figure 10 shown, the low-salinity core appears around 4°S. Is there a relationship with CSC current?

We stated that “During DJF, the currents through the Karimata Strait driven by the monsoonal winds carry waters from the SCS into the Java Sea and from the Java Sea up into the Makassar Strait (Figure 3a).” (lines 192-194 in the revised manuscript), and “The Java Sea freshening spreads into the southern Makassar Strait (Figure 2a-b), which is further re-enforced by the runoff from the southeast coast of Kalimantan” (lines 358-360 of the revised manuscript). Therefore, the low-salinity core that centered near 4°S in Figure 10 during boreal winter-spring reflects the intrusion of the low-salinity waters from the Java Sea and the effect of runoff from the southeastern side of Kalimantan. The currents that carry the Java Sea low-salinity water into the southern Makassar Strait are driven by local monsoonal winds, although they can also be considered as an extension of the South China Sea throughflow that go through the Karimata Strait.

7. I understand the difficulties of quantifying the halosteric and thermosteric contribution to SLA. How much influence does the change in meridional pressure gradient have on ITF? Can it be quantified?

ADCP measurements of currents in the Makassar Strait up to as shallow as 20 m (Figure 2 in Gordon et al. 2003, Ref.17) showed that the averaged southward velocity was approximately 0.75 m/s during boreal winter-spring but nearly zero during boreal summer. This reflects the effect of the seasonal change in meridional pressure gradient (associated with the seasonal freshwater plug) on the upper-layer ITF flow. Note that the currents in the upper 20 m were not measured due to ADCP’s limitation.

Although SSS and sea level anomaly provide a proxy for the seasonal changes of pressure gradient along the Makassar Strait, a quantitative calculation of the relation between the along-strait pressure gradient and the resultant upper-layer ITF transport requires vertical profiles of salinity and temperature measurements along the strait, which are not available. We would not attempt to use the WOA13 climatology to conduct such an investigation because of the artifacts in WOA13 that we pointed out in relation to Figure 2. The main focus of our study is the sources of the seasonal freshwater plug. Not having a quantitative estimation of the relation of meridional

pressure gradient and ITF transport does not affect the main message of our study. However, the reviewer's suggestion definitely warrants a future investigation, for instance, by conducting model sensitivity experiments or an analysis of a data assimilation product that has good fidelity in representing the vertical profiles of salinity and temperature.

Reviewer #3 (Remarks to the Author):

This study demonstrates the utility of salinity remote sensing data in the Indonesian Seas to understand the interaction between seasonal water cycle and the Indonesian Throughflow. In contrast to previous studies suggesting monsoonal freshwater input from the South China Sea through Karimata Strait, this study shows that the boreal winter-spring freshening of surface water parcels in the Makassar Strait is caused by local precipitation and run-off from Kalimantan, which leads to a reduced southward pressure gradient that would weaken the Indonesian Throughflow. I think this study meets a major challenge in oceanography for the Indonesian Seas by carefully using latest remote sensing datasets and thus recommend publication as is.

We greatly appreciate the positive comments from the reviewer.

Response un-related to the reviewers' comments:

In the original manuscript we referred to Figure 3 current vectors as surface current, but the vectors showed were actually surface current anomalies. In the revised manuscript, we have replaced the surface current anomalies in Figure 3 with surface currents. This does not affect any discussion and conclusion because the surface current anomalies and surface currents in the Java Sea are almost the same since the seasonal anomalies are much larger than the annual mean in the Java Sea.

Reviewers' comments:

Reviewer #2 (Remarks to the Author):

This is second round review. This paper investigated the freshwater plug in the Java Sea and its relationship with water cycle in the MC based on ocean-atmosphere-land satellite observations. Contrary to previous studies, this paper showed that the freshwater plug is caused by local precipitation and runoff from Kalimantan and causes a change in meridional pressure gradient along the Makassar Strait that regulate Indonesian Throughflow. I suggest the authors to consider following detailed comments to improve the manuscript before publication.

1. The major concern of this paper is the sources of freshwater plug in the Java Sea. The direct relationship between the freshwater plug and ITF seems not robust enough. According to present result, it's hard to tell that the freshwater plug is the major or secondary factor to ITF seasonality. Thus, the title seems not appropriate. Furthermore, the impact of freshwater plug on ITF does not seem to be new, which has been investigated in previous studies and mentioned in Lines 87-89 (Introduction).

2. What's impact of the monsoon winds on SLA? Is it sea surface salinity and temperature sufficient to explain the effects of salinity and temperature, although the depth is shallow in the western and central parts of the Makassar Strait, however, it is much deeper in the eastern part of the Makassar Strait. Could you discuss the SSS and SST effects in the two regions separately?

Response letter for manuscript NCOMMS-18-26394A
(second revision)

Reviewer #2 (Remarks to the Author):

This is second round review. This paper investigated the freshwater plug in the Java Sea and its relationship with water cycle in the MC based on ocean-atmosphere-land satellite observations. Contrary to previous studies, this paper showed that the freshwater plug is caused by local precipitation and runoff from Kalimantan and causes a change in meridional pressure gradient along the Makassar Strait that regulate Indonesian Throughflow. I suggest the authors to consider following detailed comments to improve the manuscript before publication.

We appreciate the time and additional comments by reviewer#2 to help improve the manuscript further.

1. The major concern of this paper is the sources of freshwater plug in the Java Sea. The direct relationship between the freshwater plug and ITF seems not robust enough. According to present result, it's hard to tell that the freshwater plug is the major or secondary factor to ITF seasonality. Thus, the title seems not appropriate. Furthermore, the impact of freshwater plug on ITF does not seem to be new, which has been investigated in previous studies and mentioned in Lines 87-89 (Introduction).

As the introduction stated and the reviewer pointed out, the impact of the seasonal freshwater plug on the seasonal ITF transport has been investigated (e.g., Ref.17). In other words, previous studies have already concluded that the seasonal freshwater plug regulates the ITF, the low-latitude chokepoint of global ocean circulation. The unique contribution of our study is the identification of the sources of the seasonal freshening (namely, MC local monsoon rain and runoff from Kalimantan). Thus, putting together our contribution that identifies the main sources of the seasonal freshwater plug and the previous result about seasonal freshwater plug regulating the ITF, the title of the paper is appropriate. Moreover, in this revision we have added a quantitative discussion about the important role of the seasonal freshening on the meridional pressure gradient along the Makassar Strait that acts to regulate the ITF flow, which also addresses comment#2 below by the reviewer.

2. What's impact of the monsoon winds on SLA? Is it sea surface salinity and temperature sufficient to explain the effects of salinity and temperature, although the depth is shallow in the western and central parts of the Makassar Strait, however, it is much deeper in the eastern part of the Makassar Strait. Could you discuss the SSS and SST effects in the two regions separately?

This is a good suggestion by the reviewer that we believe has strengthened our major results. In this revision we added the calculations and related discussion of the relative contributions of SSS and SST to meridional SLA gradient in the Makassar Strait associated with the freshwater plug. We performed such an assessment for the entire Strait as well as the eastern and western halves of the Strait separately. The results showed that the seasonal SSS variation is more than enough to explain the meridional SLA gradient for a 50-m mixed-layer depth. For a 25-m mixed-layer

depth, the seasonal SSS variation still explains more than 50% of the meridional SLA gradient in the Strait. SST variation has little effect on the meridional SLA gradient.

These discussions are shown by the track changes in lines 432-441 in the “Downstream effects on the Makassar Strait” section (replacing an old paragraph that is now redundant given our explicit calculation), and lines 648-678 in the “Method” section (under a new sub-section “SSS and SST contributions to SLA gradient along the Makassar Strait”).

REVIEWERS' COMMENTS:

Reviewer #2 (Remarks to the Author):

I appreciate that the authors have made a great effort to revise the manuscript, which shows interesting and exciting results based on latest remote sensing datasets. This paper is well-written and well-structured, and the results are attractive and novel. I recommend to accept it.

Here I only have a few suggestions to authors to consider in their study.

Due to data limitations, the residual item in the upper ocean is positive and large. I understand the difference in observation methods/tools are major cause. Even so, does any dynamics processes involve in?

The data used in Ref. 17 is 20 years ago, and it might be better to update the data, since the co-authors had made a great progress in field observations in this region.

Response to final comments by reviewer#2

Reviewer #2 (Remarks to the Author):

I appreciate that the authors have made a great effort to revise the manuscript, which shows interesting and exciting results based on latest remote sensing datasets. This paper is well-written and well-structured, and the results are attractive and novel. I recommend to accept it.

We thank reviewer#2 for the positive comments, as well as the constructive suggestions and time spent through the revisions.

Here I only have a few suggestions to authors to consider in their study.

Due to data limitations, the residual item in the upper ocean is positive and large. I understand the difference in observation methods/tools are major cause. Even so, does any dynamics processes involve in?

We had addressed these in the original version of the paper. The positive residual term is not necessarily due to data limitation. And yes, ocean dynamics such as vertical mixing does contribute to the residual term (lines 320-322 of the final revision). The residual term being positive is consistent with the effects of vertical mixing and evaporation, both trying to counteract the freshening effects due to precipitation and horizontal advection (i.e., counteract the negative salinity tendencies). This was explained in lines 342-354 of the final revision: *“The effects of -P and horizontal advection generally assist the seasonal variation of SSS tendency while the residual term generally counteracts the combined effects of -P and horizontal advection, especially during DJF and MAM. The counteracting effect of the residual can be understood as the dissipative influence of vertical mixing (and evaporation to a lesser extent) that attenuate the upper-layer freshening.”*

The data used in Ref. 17 is 20 years ago, and it might be better to update the data, since the co-authors had made a great progress in field observations in this region.

This is a rhetorical comment about the in-situ mooring data used Ref17, not about the satellite data analyzed in our study. Ref20 provided updated mooring data record in the Makassar Strait through the 2000s. Additional mooring record for the past several years will be published by co-author A. Gordon in separate papers.